# Minimum Space When Transporting Pigs: Where Is the “Good” Law?

**DOI:** 10.3390/ani14182732

**Published:** 2024-09-21

**Authors:** Terry L. Whiting

**Affiliations:** Government of Manitoba, Winnipeg, MB R3B 3M9, Canada; terrywhiting9@gmail.com; Tel.: +1-204-795-7875

**Keywords:** stocking density, humane pig transport, performance-based regulation, pork supply chain, pig welfare, animal cruelty, industrial farming

## Abstract

**Simple Summary:**

One may think that the number of pigs that can fit on a truck is a simple scientific question. It is not. It is vitally important that scientists can describe their work simply and concisely to the public, especially when that work is providing the basis for regulation in the public interest. Successful regulation of human behavior requires the regulated to agree with the rule. Commercial livestock transporters have an interest in loading a livestock compartment to the maximum to minimize costs. Livestock production academics, veterinarians, and animal welfare activists have been working for decades to determine the level of livestock crowding in transport containers that would be an appropriate threshold for regulatory enforcement. To date, there is no real consensus on this issue across species or animal size within a species. The EU countries agreed to a maximum floor pressure for market pigs of around 110 kg body weight to be 235 kgm^−2^ in 2005, while North American regulators have no legal standard. Current practice in North America allows for significant crowding of pigs in excess of the EU standard. Using the pig as an example, this paper examines the practical barriers that have for decades prevented emergence of a consensus on the “Is this truck full?” question.

**Abstract:**

This paper focuses on the problem of numeracy when writing regulations, specifically how to describe a threshold for crowding of pigs during transport, considering transported pigs range in body mass from 5 to 500 kg. When scientific findings provide the basis for regulation in the public interest, those findings must be communicated in a consistent way to regulators and policymaking bodies. Numeracy is the ability to understand, reason with, and apply appropriate numerical concepts to real-world questions. Scientific understanding is almost always based on rational understanding of numerical information, numeracy. The threshold of administrative offenses is often a numerical description. Commercial livestock transporters have an interest in loading livestock compartments to the maximum to achieve the largest payload allowed by axle weight laws, as is the case in all bulk commodity transport. Maximizing payload minimizes costs and environmental hazards of fuel exhaust and can benefit the public with lower pork prices, but has a serious animal welfare risk. Livestock production academics, veterinarians, and animal welfare activists have been working for decades to determine the level of livestock crowding in transport containers that would be appropriate for regulatory enforcement. The scientific discourse has been plagued by a lack of numerical standardization when describing results of trials and forming recommendations. Exceeding specific numerical thresholds is the core to implementing enforcement actions. This paper examines the communication and other barriers that have prevented emergence of a consensus on this question and provides a direction toward resolution. Further confirmation of effects of crowding livestock in transit is needed. This paper suggests that articulating an enforceable standard in pig transport is possible. In inspection for compliance, discovering the LP_50_ (lethal pressure—50) for slaughter-weight pigs is an initial global benchmark goal. The LP_50_ is the loading floor pressure in a commercial transport compartment, under field conditions, that would result in the death of at least one pig in the group 50% of the time.

## 1. Introduction

In 2006, the World Organization for Animal Health (WOAH-OIE) recommended animal transport times be kept to a minimum with sufficient space allowance for animals to lie down during transport, with consideration given for climate and ventilation capacity of transport vehicles [1] (now Chapter 7.3, Terrestrial Code). All the WOAH considerations—time, space, temperature, humidity—have a numerical measure, but no current threshold limits. As a signatory to the OIE, Canadian scientists in 2008 believed that global livestock welfare transport standards would soon follow the 2006 international agreement [2]. Almost two decades later, there still is no international consensus of the minimum space that should be allocated to a pig in transport. This paper focuses on this one condition of humane transport, the two-dimensional space required to humanely transport a pig.

The option for a pig to lie down while in transit has a normative power with the common voter and should be a minimum legal requirement in countries that recognize a responsibility for humane care of animals exploited for human ends, in this case pork. The question of what this space requirement is has an objective solution via scientific inquiry. Remarkably, this remains an open question. This paper will review and identify possible causes of lack of scientific progress and propose a possible pathway to consensus.

All workers in swine welfare and meat quality agree that transportation is a serious stress on the animal and a cause for concern. Market-weight pigs arriving at the abattoir dead or recumbent due to fatigue are easily identified, and there is general agreement that downer and dead pigs have suffered during transit [3]. An animal care inspection and enforcement program should be triggered when there are more than one dead or disabled pigs in a single compartment of a commercial carrier on arrival at a control point. This paper will suggest that data collection at unloading can largely be used as evidence of violation of a legal prohibition of overcrowding; by comparison with the same load compartments without death loss. Over time, accumulated individual compartment data could also be used to titrate crowding against death loss to establish a legitimate numerical measure of lethal overcrowding. Ongoing monitoring would confirm a performance-based numerical standard acceptable to the regulated, the regulator, and the concerned public.

The purpose of this report is to present the question of the humane carrying capacity of livestock vehicles commonly hauling pigs from a different perspective. How can production-facing animal scientists describe the limits of carrying capacity in a way that allows for fair and effective policing by officers with no knowledge of livestock or their physiology? There is currently both a gap in knowledge reflected in a lack of a clear loading threshold for all weights of pigs and a lack in agreement on how that threshold can be presented to the regulated, the inspectorate, and the judiciary.

## 2. Good Law

A “good” law articulating a social policy is intended to bring about behavioral change in the group of people targeted. Regulators intervene where there is a public good to achieve and where the individual may have cause to do otherwise. Regulatory compliance is most successfully encouraged where the rules are 1. intuitively directs to a positive outcome, 2. the behavior is specific and objectively measurable, 3. the behavior is voluntary modifiable, 4. the requirement is not culturally specific, and 5. the behavior is scalable and not prohibited. Rules must be directed at a common good, make common sense, and be easy to describe and enforce, and for breaking the law to be intuitively associated with a negative outcome for society. For laws to be effective, the regulated must agree to be compliant. The large question in regulation is not why people break the law, but why they obey [4].

In the last two decades, the emergent popularity of performance-based regulation could be added as a sixth requirement of good regulatory law. Performance-based regulation is characterized by identification of a specific outcome and not on the method of achieving that outcome [5]. A prescriptive animal transport law would be “When Equidae are transported in a closed container, the distance from the floor to the ceiling must not be less than 1.85 m.” The performance-based equivalent may be “When Equidae are transported in a closed container, an animal’s head must not be in contact with the ceiling when the animal is standing in a natural position.” Rules must be specific such that an assessment of non-compliance and what specifically this non-compliance is based upon must be described in the writing of the offense. Commitment to performance-based regulations is largely ideological, as there is no convincing evidence that performance-based regulations are a solution to poorly written regulations [6].

The rule “It is an offense to exceed the posted speed limit” meets all the aforementioned five requirements. This law is so reasonable, so easy to administer, and so flexible to local conditions, being culturally independent and voluntarily compliable, that it has been adopted worldwide. Due to its specificity, regulating speed limits is often misconstrued as prescriptive law. This law is purely performance-based, as any method of achieving the performance goal of “below posted speed limit” is equally acceptable. Speed maximums have regulatory “precision” (transparency, accessibility, and congruence) that guides the wording of legislation [7]. In addition, enforcement is extremely easy, because an infraction can be objectively and legitimately measured by any one of several technologies. It is a legitimate rule, as the regulated, the policing, and the courts all see this issue from the same perspective [8].

A good law requires a method of measuring human behavior against the behavior prohibited in the rule. For live animals in transit, mortality, dead on arrival, and dead in pen awaiting slaughter, are obvious performance-based indicators, but are neither a sensitive nor specific measurement of poor animal welfare. However, not all mortality in transit can be avoided or is necessarily due to poor welfare, as a few animals within the production system will die within the transport time, even if they have not left the farm. Around 5% of pigs entering the 120-day grow–finish phase will die [9]. Although livestock overcrowding in transit is nominally illegal under national statute in Canada, prosecution is rare.

## 3. Language and Units: Describing Space Allowance

In the scientific literature, the language used to describe the space available to livestock in transit has several options, most commonly “space allowance” and/or “stocking density.” Stocking density as used is a misapplication of the meaning of density. Density refers to weight related to volume, a three-dimensional construct, whereas research on loading pigs universally refers to floor space, a two-dimensional measurement. In making recommendations for live fish in transport [10] and fish husbandry [11], density is used in the correct way. The mathematical units used to describe the animal space in livestock transport also vary and reflect the way particular authors construct the question of animal welfare. Units such as area per animal (m^2^/animal) [12,13], weight of animal per area (mass/m^2^) [14], or standardized space unit (m^2^/100 kg) [15,16] have been used to describe the safe increase in deck pressure for animals in transit as the animals increase in size. This paper has followed the pressure measurement unit of kilograms per square meter to describe threshold, as this is the convention used in EU legislation [17], the CARC recommendation [18], has functional superiority, and is a personal preference. 

Providing graphical representation of numerical values is a superior method of legal and scientific communication when compared to the limited utility of recommended weight–space tables (Figure 1). Converting the tabular recommendations into a graphic of bodyweight (kg) vs deck pressure (kgm^−2^) reveals that as pigs grow there are significant gains in the efficiency of transport deck space use. While crowding from the growing pig perspective is constant, floor pressure can increase dramatically up to a maximum of 150 kg body weight, after which heavier pigs, cull boars and sows do not tolerate increasing deck pressure with increasing body weight. 

There is an objective mathematical relationship between pig body mass and minimal tolerable floor space allowance as a pig grows from 5 to 140 kg. Numeracy is the ability to understand and work with numbers and is essential for understanding the biological needs of pigs and law enforcement. Numeracy includes the intuitive understanding of simple graphical representations of numerical descriptions of reality. The same space allowance recommendations graphed as space per animal and floor pressure can be presented on the same graph using a single X and two Y axes, (Figure 2). There has been limited work on minimum compartment height [20], and until such research is completed, it would be an error to refer to space allowance for animals other than in terms of area.

In a recent EFSA scientific opinion [13], the panel chose various narrative and numerical ways to describe space allowance. The panel differentiated horizontal space (space allowance) from vertical space (headroom), recognizing but not addressing the erroneous use of stocking density in transit. The opinion authors eschewed the use of floor pressure units, not acknowledging that kilograms per square meter was the regulatory unit chosen by the EC [17]. In articulating a recommendation, the panel endorsed the allometric equation of Area = 0.027 × Weight^2/3^ for all pigs in a transport compartment to be able to lie in a semi-recumbent posture [13] (p. 88). In addition to the high-resolution, inclusive weight range allometric equation recommending minimum space allowance, there is an adjacent table (Table 29) in the document, a weight–space table, providing four specific body weight increments and recommended area. This is an odd manifestation in a scientific document where both a fine–broad and a very coarse–limited recommendation is given to the same question. The EFSA [13] recommends (at Table 29, p. 88) 202 kgm^−2^ (160 kg/0.79 m^2^) for a 160 kg pig group load, significantly less crowding than is common in North American practice. In the CARC model of space allowance for pigs in transit, at about 140 kg pig body weight, the deck pressure reaches a maximum of 280 kgm^−2^ (Figure 2) [18].

## 4. Performance-Based Regulation

Performance-based regulation (PBR) emerged in recent decades when attempts to regulate complex human endeavors such as producing chemicals and protecting the environment, assuring worker safety in inherently dangerous workplaces, and the safety of complex transport systems failed [22]. Performance-based regulation is predicated on the notion that regulation should focus on achievement of regulatory objectives and leave it to the regulated entities to determine how best to achieve them. Many regulatory administrations believe (or have been directed to act like they believe) that this method of operation will result in the best possible world [23]. An early influential evangelist for the PBR initiative was the OECD [24]. The limited empirical evidence that PBRs are inherently superior to other regulatory options has not diminished the enthusiasm for the approach [22,25]. The performance of the regulator is notoriously difficult to objectively evaluate [25], which provides a fertile ground for thriving ideologies.

The control of diesel engine emissions by standardized exhaust testing is an exemplar of PBR. Standardized exhaust testing displays two of the core priorities of performance-based standards [5]. First, exhaust testing has high specificity: it is a tight standard, as opposed to a loose standard. A PBS addressing a technically complicated system requires targets to be articulated with a high level of mechanical engineering-based literacy. Automotive engineers have a clear target and a long chain of causation, providing multiple intervention options to minimize the environmental harm in exhaust composition, leaving space for innovation. The exhaust test also has proximity to the regulatory goal, that of environmental protection. By comparison, the prohibition of tetra-ethyl lead in vehicle fuel was a prescriptive regulatory approach to control the environmental risk of lead with clear advantages over performance-based regulation, the measuring of lead in vehicle exhaust. Although the exhaust emission PBS is widely adopted and generally effective, it can be circumvented in part due to inherent complexity and the nature of the regulated parties [26].

In theory, PBS allow firms to select the most effective or lowest-cost option to achieve compliance; however, in some regulated activities, there is only one imaginable option. In the prohibition of vehicular speeding, an outcome standard (PBS), voluntary slowing of the vehicle by the operator is the most obvious method of compliance, but innovation is evident. Many North American commercial trucking companies, sensitive to the combined probability of detection and the escalating nature of sanction, have installed after-market governors in fleet vehicles to assure company compliance with speed limits, and speed governors in heavy trucks are compulsory in some jurisdictions [27]. With the innovation of smart automobiles and GIS technology, in the near future, it may become impossible to speed in a school zone, a techno-regulatory innovation likely to receive public support.

After over 25 years of industry consultation, the humane transportation regulations, Part XII of the Canadian Health of Animals Regulations [28], were revised (Table 1). The final agreement between the regulator and the regulated was guided by a strong preemptive commitment to performance-based regulation. In reviewing the amended prohibition of overcrowding, it is difficult to identify how the conditions of pigs transported in Canada improved in February 2020. With the new definition of “overcrowding,” the offense that must be proven at trial is no clearer than the previous definition. The regulator chose not to articulate clear, transparent, numerically enforceable outcomes. Having vague outcomes is not a necessary or desirable component of performance-based regulation.

## 5. Trucking: A Highly Regulated Enterprise

Trucking of livestock is a competitive commercial enterprise and trucking companies need to make a profit. This desire for a profit requires economical movement of large numbers of animals (weight) long distances with minimized costs in salaries, machinery usage, and time. The trucker, unconstrained by good judgment, would load as much as possible and drive as fast as possible, an example of uncontrolled capitalism posing a public threat. Heavy trucks generate multiple public hazards, such as road safety (multiple domains), infrastructure wear (roads and bridges), environmental pollution, and for our purposes, animal welfare. Animal care regulations in livestock transport by road operate in a matrix of regulatory oversight, with a hierarchy of compliance behavior based on probability of detection and certainty of sanction. The matrix includes maximal dimensions of trailers, maximal axle weights, maximal hours of service restrictions for drivers, and maximal confinement times for animals. These numerical maximums are created by different regulatory instruments and created without significant consideration of rule conflict. The time limitations for truck operators and the limitation of time in confinement for animals are interdependent. Trucking as an industry is by nature a dangerous workplace, with poor pay and high turnover, most prevalent in long-distance transport [29], requiring continual training. Inexperience and lack of driver training may be an additional structural welfare hazard for livestock in transport [30].

The concern for human health and safety, infrastructure protection, and environmental risks has driven the axle weight laws, speed limits, maximal hours of operation laws, and the maximal dimensions of vehicles. Animal protection is a specific latecomer to the rulemaking table, affecting a very small part of the trucking community. Maximal trailer dimensions are controlled at manufacturing and enforced at licensing. Sub-national transportation authorities also comprehensively enforce the road axle weight restrictions with significant fines and inconvenience imposed for violations. The axle weight laws and the animal overcrowding law (if one exists) have a presumed hierarchy of compliance based on probability of detection and severity of sanction. Administrative law is based on the probability of detection and severity of sanction regulatory enforcement model (deterrence). Vehicle axle weight restrictions command compliance primacy, as frequent inspection and enforcement of this feature is assured. The adoption of weigh-in-motion technology by enforcement agencies [31] may further assure compliance with this directive and for it to be dominant over regulation preventing livestock overcrowding, where detection is unlikely unless there is animal death in transit.

Rigorous regulation of commercial vehicles is a feature of livestock transport in North America (Figure 3) and Europe [32]. For Canada–US livestock transport, axle weight compliance rules intersect with the practice of placing groups of animals in trailer compartments, creating uneven utilization of available floor space in a trailer and a high risk of noncompliance for some equipment (see case study below).

The specific challenges for swine transport in Canada, including equipment available, have been recently reviewed [33]. In Canada, hog-specific trailers and dual-use cattle–hog trailers have three levels of flooring in the middle section of the trailer when hauling pigs, a “potbelly trailer.” Two axles (tandem) are interconnected so that both axles bear the same load, whether drive axles or trailer axles. The three axles on tri-axle trailers similarly share the load for the purpose of axle weight compliance. Pig-specific, three-level commercial trailers are 14.6 m (48 feet) in length, with 37.5 m of running deck. The curb weight of an empty tri-axle trailer is slightly greater due to the weight of one additional axle and increased aluminum in the box of the longer trailer. The maximum trailer length is 16.2 m (53 feet), most having three rear axles.

**Figure 3 animals-14-02732-f003:**
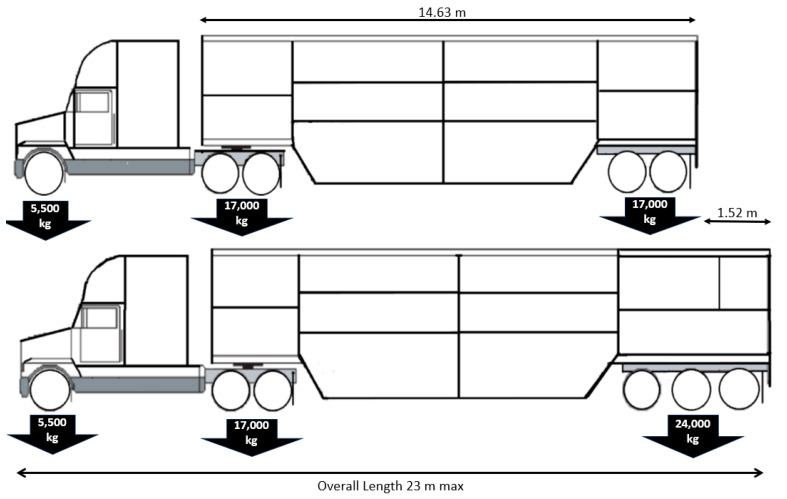
Schematic representation of the internal conformation of two common livestock trailers used for transporting slaughter pigs and cull sows and boars. The maximum gross vehicle weight and dimensions (Manitoba) are measured at three points: the steering axle (5500 kg), the combined tandem drive axles (17,000 kg), and the rear axles (17,000 kg, 2-axle, 24,000 kg, 3-axle) [34]. Vehicles with these dimensions and loaded axle weights can generally drive anywhere in Canada and the US.

In Figure 3, the 14.3 m trailer has roughly 95 m^2^ of deck for pigs, and the horizontal middle section of the center deck is removable for cattle. The extra length in the 16.2 m tri-axle gives two extra deck areas of 2.5 by 1.5 m or a total of 7.5 m^2^. This increase in pig deck space of 8% is accompanied by almost 7000 kg increased weight allowance, a 17% increase in axle weight allowance. The intersection of axle weight allowances and the requirement of market pigs for space allow the rear compartments of 16.2 m tri-axle trailers to be severely overcrowded and remain within axle weight allowances.

## 6. Transporting Slaughter-Weight Pigs by Road: Welfare Assessment

The welfare of slaughter pigs in transit has been measured by mortality, carcass bruising, serological measurements of physiological stress, and behavioral observations. Warriss (1998) addressed the question of space allowance with a distinctly intuitive approach. The study involved adding similar pigs to a fixed-area container and viewing the group from above: when no floor could be seen, the pen was “full” [35]. In this observational approach, it appeared that a floor pressure of 250 kgm^−2^ allowed sufficient space for a group of 100 kg market pigs to lie down in sternal recumbency. This general approach to identify the space occupied by a pig continues with computer-assisted image analysis [36,37].

Other groups were interested in the possible cost of meat quality defects due to crowding or increased mortality. Gade and Christensen, also in 1998, evaluated crowding by transporting slaughter pigs a short distance, transport time less than 2 h, decreasing floor space per 100 kg pig in four stages (200, 238, 265 and 285 kgm^−2^), and found that pork quality appears to be a relatively insensitive measurement of animal welfare and crowding during brief transport. Pork quality does not appear to be dramatically affected by a total trip of 2.5 h at densities up to 285 kgm^−2^ [38]. Lambooy and Engel (1991) worked with slightly heavier 110 kg pigs at 186, 232, and 278 kgm^−2^ and longer distances of 25 h, and were in general agreement [39]. They found that in compartments with a floor pressure of 278 kgm^−2^, not all animals were able to lie down at the same time, agreeing with the Warris finding, and went on to recommend that compartment floor pressure for moving slaughter-weight pigs to market should not exceed 232 kgm^−2^ for animal welfare and meat quality reasons [39].

Using commercial animals and accepted practices for transporting market-weight pigs (131 kg) three hours to slaughter in spring and fall, Ritter et al. examined handling and transport challenges, including space allowance in the trailer. Alternating floor pressure per compartment was accomplished by adding a single pig to the group, and allowed titration of increasing floor pressures of 252, 268, 283, 299, 316, and 358 kgm^−2^ against the frequency of death loss in transit and the frequency of stressed or fatigued pigs on arrival [40]. These floor pressures all exceeded the maximum recommendation by Warriss of 250 kgm^−2^. At the time, the National Institute of Animal Agriculture recommended a 302 kgm^−2^ limit [41], but the six floor pressures used in the study were common. The results of the study suggested that death and debility of transported market pigs increased at about 283 kgm^−2^. This titration study was essentially repeated in the same production system using slightly lower floor pressures (240–289 kgm^−2^) and slightly lighter pigs (125 kg), and found no effect of crowding on death loss. Only two of the six challenges, 301 and 339 kgm^−2^, were more than 300 kgm^−2^ [16].

Although details varied, it appeared that there was a general consensus that pigs could be crowded to a state of “standing room only” for short trips where standing is preferred by the pig or that pigs around 100 kg could be loaded with the intent of allowing sufficient space for the group to lie down on long trips. Naturally, the floor pressure (kgm^−2^) is higher with standing room only than in situations where being recumbent is possible for all. More recent work confirms this fact [37,42].

The concurrent rapid intensification of pig farming initiated similar space requirement questions in the efficient or maximal use of barn pen space as pigs grew from weaning to market weight and in the housing of sows in gestation crates. Solutions to these questions recruited scaling theory. It is widely accepted that as a pig grows, their weight changes as a cubic function, but the space floor space (or shadow cast) increases as a squared function, leading to the general predictive allometric scaling law A = (k)W^2/3^, where A is area in m^2^, W is liveweight in kg, k is a species and condition-specific value such as standing room only or space to lie, and the scaling exponent is two-thirds, in predicting space requirement for livestock. Allometric power laws of this type are pervasive in the study of animal physiology. An early example is Kleiber’s law, which predicts metabolic rate from animal body size where the scaling exponent is three-quarters, where k values vary [43]. The scientific evidence of allometric scaling is broad, for example, the cross-sectional areas of mammalian aorta and tree trunk scale are predicted by the Mass^3/4^ power law [44]. The symbol k generated from research data is used by convention in humane livestock transport research [45] and in setting standards for pen capacity in barns [46]. In both barns and transport, safe floor pressure for small pigs is much lower than safe floor pressure for large pigs, and the relationship is not a straight line.

By 2009, it was generally accepted that the minimum area per pig in transport (standing room only) was approximated by a k value of 0.020 and for simultaneous group sternal recumbency by a k value of 0.027 [47]. This recommendation for transport has been retained in current EU opinion as standing-room-only pigs in transit is estimated by a k value of 0.027 [13] (p. 88), which is 0.62 m^2^ per animal and 177 kgm^−2^ for a 110 kg market pig (Figure 4). However, the 2005 regulation permits floor pressure up to 235 kgm^−2^ for 110 kg market-weight pigs [17] (p. 31).

At the new millennium, it appeared that the scientific community had made a good start on finding an empirical scientific numerical consensus on the question of how many pigs of weight X will fit on a truck with deck area Y. However, in 2024, the North American regulators are silent and current European legislation has not significantly advanced or confirmed the maximal crowding standard in the 2005 agreement on a single point, a floor pressure of 235 kgm^−2^ for a market-weight pig in the 100 kg body weight range [17,37]. In comparison to North American standards, the 2005 EU agreement may be needlessly conservative (Figure 4).

## 7. Communicating the Concept of Crowding Pigs

As animals increase in size, their weight increases as a function of their length^3^, as weight is a function of volume (three dimensions), while the floor space they cover increases as a function of their length^2^ (two dimensions). Figure 4 illustrates two mental constructs of how pigs grow and how increasing body weight affects efficient floor utilization in transport. The standard allometric formula reflected in the current EFSA recommendation [13] (k = 0.027) envisages a pig that grows like an expanding sphere and that geriatric pigs are scale images of 28-day-old pigs. The CARC and NPC recommendations view a pig as being largely an unchanging scale model from weaning to standard slaughter weight at around 140 kg [48]. In this pig growth model, further skeletal growth in breeding animals above 140 kg is not characterized by maintaining scale body conformity to the market pig, and there is not increasing efficiency of use of deck area in transit by breeding stock. When graphically represented, the relationship between body mass and maximum deck pressure becomes a constant ratio at weights above 140 kg.

Trucking efficiency, measured as maximum weight by area of deck, peaks at around 130–150 kg body weight. This weight includes both the largest intended slaughter-weight pigs (in Canada, the current target slaughter weight is 136 kg) and the lightest culled first-litter gilts in poor body condition. Cull sows and market pigs of the same body weight can have very different appearances. After 140 kg BW, pigs continue to require more floor space as they continue to grow, but the efficiency of use of trailer floor space does not continue to improve. In Manitoba, a payload of cull sows for export is seldom significantly heavier than a payload of market hogs using the same trailer, contrary to the A = k(W^2/3^) theorem. However, this may be an artifact of compliance with local axle weight laws of the 14.6 m trailer conformation. This trailer, with deck space of 95 m^2^ loaded at 280 kgm^−2^, has a 26,600 kg payload, leaving 12,600 kg for the weight of the tractor and trailer to remain in weight compliance. Empty tractor–trailer units are between 12,000 and 15,000 kg. For the 14.6 m trailer combination (Figure 3, top), local axle weight limitations may in effect decrease the risk for slaughter-weight pig overcrowding using this particular trailer.

The CARC-NPB model of pig growth supports a maximum recommended floor pressure value somewhere around 280 kgm^−2^ for all pigs. This model originated with the Canadian swine trucking industry, which exports millions of weaner pigs and most cull sows and boars to the United States, where there is veterinary border inspection for humane transportation. The two approaches articulated graphically in Figure 4 also differ in cognitive origins. The EFSA allometric approach originates with the space an individual pig needs [36], whereas the CARC approach is group of pigs in origin, i.e., how much trailer a mass of live pigs requires with the knowledge that it varies dramatically with average pig weight. The standard articulated by the NPB [19] is remarkably congruent with the CARC standard [18] (Figure 4, red and yellow lines). After 125 kg, the NPB line loses the internal agreement present in the first four weight categories.

The NPB and CARC agree that at some body weight, there is a maximum floor pressure that should not be exceeded in considering the transport of cull sows and boars. There is physiological evidence to support this belief that was not available at the time the original recommendations were made. As pigs grow, cardiac function measured by stroke volume and cardiac output maintains proportion to body weight and scales up until pigs weigh about 150 kg [49]. It is clear that proportional to body weight, mature sows’ cardiac performance is disproportionally low for body weight [50,51]. This may explain the field observation that cull boars exported to the US are very sensitive to the rigors of land transport in Canada compared to other cull types.

## 8. Case Report

In August, prior to a 2020 regulatory amendment, a load of hogs was assembled at Lloydminster, Alberta and consigned to slaughter in Edmonton, directly west, 249 km by high-volume road, about 2 h 35 min. When arriving at the destination, the load was rerouted to Winnipeg, Manitoba, a further 1305 km, about 13 h 20 min. Air temperature ranged from a low overnight of 17 °C to a high on arrival of 24 °C. Relative humidity was always less than 65%. When the load was inspected at unloading, there were eight pigs dead and no distressed pigs. The author was working as an animal welfare protection officer, and this information was captured and recorded with the intent of presenting it as evidence in prosecution. This section reflects standard inspection procedures.

There were 242 (122 kg) pigs loaded, and 234 arrived alive for slaughter. Recording the location of the live and dead pigs and measuring the compartment lengths allowed the calculation of floor pressure per compartment (Figure 5). Seven of the eight pig deaths in transit occurred in the lower rear compartment, which had the highest deck (floor) pressure. The compartment with the highest deck pressure that was not associated with death loss was the upper rear compartment, loaded with 24 pigs at 307 kgm^−2^. This compartment is L-shaped to allow for ramp storage, which is integral to the trailer design, and provides more perforated wall space per pig than other compartments. Rear compartments are recognized as the best ventilated in passively ventilated livestock trailers [13] (p. 48). The nose compartments are the lightest loaded in this tractor–trailer combination as expected, as there is a risk of tractor steering axle weight violation if the center of gravity of the transport unit as a whole is too far forward.

In review of this incident with the abattoir veterinary staff, it was concluded that the absence of pigs in distress indicated that the crowded pigs died early in the trip, allowing time for previously fatigued pigs to recover. Six of the eight compartments were in excess of the CARC recommended maximum of 280 kgm^−2^. It was noted that the ambient temperature was cool and dry for August in the Canadian prairies, the majority of the trip was overnight, and the vehicle was unloaded at the start of the morning shift, 6:00 a.m. The dead pig in the lower middle compartment was considered a random death.

This incident is an example that has been repeated. It is hypothesized that pigs on trips more than 4 h will prefer to lie down. When there is insufficient space to lie down, the pigs physically compete for area. At this body weight, pigs will not permit other pigs to lie atop resting pigs, so there is little sharing of area. The increased physical activity results in fatigued pigs [52], and some go on to die. It is further assumed that pigs resting on a carcass will not continue to fight for floor area and can recover with time.

## 9. Discussion

Many of the references cited in this paper are more than 20 years old. Current university animal use ethics committees would probably balk at an experimental design where pig death or the clinical manifestation of a stressed/fatigued pig (open mouth breathing, muscle tremor, reluctance to walk, involuntary recumbency, blotchy skin, and hyperthermia) [52] was the outcome measure sought. Sublethal overcrowding may not affect pork quality significantly and thus may be a “necessary” animal discomfort in the matrix, where we balance animal well-being against stakeholder profits in postmodern capitalist ethical debate. Recent work in Europe confirms that if EC 1/2005 is complied with (235 kgm^−2^ maximum pressure), floor space allowance for slaughter-weight pigs is not a factor in the risk of in-transit pig death, even in what are considered higher-risk, heavy phenotypes [53]. In an Italian study, there was very high compliance with the floor pressure limit of 235 kgm^−2^, with only 5 of 307 loads found not to be in compliance. The maximal crowding documented in this report had a floor pressure of only 303 kgm^−2^ [54]. The EU’s space allowance appears generous compared to mainstream pig transport practices in North America.

In application of the Canadian prohibition of overcrowding Sec 140 to the case described above, there is a significant lack of precision in the law [7]. It is challenging to achieve a conviction where the offense is created by the narrative “No person shall transport or cause to be transported any animal in any … container that is crowded to such an extent as to be likely to cause injury or undue suffering to any animal therein.” In the case example, since the offense is “overcrowding,” compartments where death had not occurred were compared with the compartment with death loss, all pigs sharing other aspects of the trip in common. The judge was convinced in the absence of an alternative explanation by the defense that an overcrowding offense had occurred.

In the case described, the judgment turned on the “likely to cause injury” phrase in the offense. In application of the law, the group loading the pigs for a three-hour tour did not meet the standard of “likely to cause injury.” The dispatcher who changed the destination, considering the clause cause to be transported, and the driver who transported were found legally liable because the 16 h trip was recognized as likely meeting the strict liability standard of the regulation. It is doubtful in the case described that conviction would have been possible without the floor pressure information from the other compartments in the same vehicle where there was little or no death loss.

In the current Canadian Section 148, frostbite (a swine-only issue) is prohibited in the overcrowding section of the regulation. A charge would have to be worded such that overcrowding was the primary cause in the chain of events allowing the development of frostbite (sub c). It is a Canadian belief (also held by the author) that overcrowding market pigs in winter can function to hold least dominant pigs in “cold spots” in a trailer. Submissive pigs will not cause dominant pigs to take a turn in the cold spot when floor space is at a premium. The problem with the “frostbite” provision in the overcrowding section of a law is that no prosecutor could possibly be aware of this indirect and unproven line of causation. In critical review of causation, a good legislative rule is located very close to the negative outcome in the causal chain of events [22]. This characteristic of causation has also been described as the proximity between the legal command and the regulatory goal [5]. In comparison, following the logic of “speed kills,” restricted vehicle speed limits in school zones reflect a very short casual chain between the command and the goal of avoiding pedestrian child vehicular manslaughter.

The offense related to animals “trampled” on trailer Sec 148 (sub b), could be reasonably prosecuted, unless the compartment was so crowded that all individual animals became recumbent and were prevented from trampling and died of postural suffocation (horses), with little evidence of trampling at postmortem evaluation. In drafting a performance-based regulation, prohibition of human behavior likely to cause animal death or serious injury is a rather insensitive performance outcome. Prevention of death in transit does not suitably encompass the regulatory goal of assuring animal welfare in transport. The prevention of suffering prior to death is also a clear regulatory goal.

An example of a draft regulation more in line with legislative goals and a tightly specified performance standard would be:


*It is an offense to load any pig for transport or to transport a pig where the resulting floor pressure exceeds the maximal deck pressure graph (Figure 1, in this document); fines double when ambient temperature exceeds 25 °C.*


In the recent EFSA Welfare of Pigs during Transport review, the recommendation of a numerical standard by formula and then repeating it in a much coarser table suggests that there is a problem of numeracy [55,56]. For this decision to be rational, the authors must believe the maximal deck pressure graphic in Figure 1 to be inherently incomprehensible by the average reader. The authors may correctly anticipate that a target audience cannot understand the allometric calculation as a mathematic description of reality, but the target audience may be able to understand graphical representation of the same standard. It is possible that policymakers and/or regulators do not understand graphical representations of numerical data. It is unlikely that there is a numeracy problem preventing agreement within the experts themselves. Graphical representation of scientific information and concepts is used elsewhere in the EFSA document, suggesting that graphical representation is not an issue internal to the authors. A remaining hypothesis is that there is a problem communicating quantitative reasoning between scientists and lawmakers, with lawmakers unable to understand or unwilling to deviate from the legal culture of presenting commands involving numerical information in tables.

Commercial truck operators are assigned ISCO-08 code (8332, Heavy Truck and Lorry Drivers) and require Level 1 Literacy and Level 2 Numeracy [32]. Tasks at numeracy level 2 require the driver to *identify and act on mathematical information and ideas embedded in a range of common contexts where the mathematical content is fairly explicit or visual with relatively few distractors. Tasks tend to require the application of two or more steps or processes involving calculation with whole numbers and common decimals, percentages and fractions; simple measurement and spatial representation; estimation; and interpretation of relatively simple data and statistics in texts, tables and graphs* [57]. For a truck driver to understand a maximum deck pressure graphic (Figure 1), they would need to have the capacity to read the pressure threshold from the average weight of the pigs loaded and understand what floor pressure means. Future research could establish that current truck and lorry drivers are unable to interpret or learn how to implement the two-step process inherent in reading a single-line graphical standard. If a law is not comprehendible to the regulatory target, the regulation will have failed. If the regulated target can quickly grasp an efficient graphical representation of a maximal floor pressure standard and the legal and policy infrastructure would fail to understand the same graphical representation, then we have a serious problem in regulating this aspect of animal welfare in transit.

In the recent Canadian consultation, specifically the goal of prohibiting the overcrowding of pigs, the Canadian legislature had agreement from the science community (CARC) and the pork producers (NPC) of what the current industry practice is, described by their own transport quality assurance program, but failed to describe a clear numerical regulatory violation in relation to overcrowding. The problem may be one of innumeracy, not in the regulatory target, but a lack of ability to reason with numbers within the regulatory community [58,59]. Alternatively, it represents a failure to innovate in adopting smart regulations.

The progress of science requires specific definitions, including units of measure. For the scientific community evaluating livestock crowding, referring to density when we intend pressure is like conflating speed and acceleration in the discourse of Newtonian physics and needs to be addressed by authors and journal editors. Adopting the standard unit of pressure (weight/area) of a loaded trailer is preferrable, especially to the end user, an individual on a specific day, loading a specific size of pig, balancing the center of gravity of a specific tractor–trailer unit to maintain compliance with a specific sub-national axle weight law. Incidentally, in Canada and the US, where imperial weights and measures dominate commerce in livestock, 300 kgm^−2^ is equal to 61 lbft^−2^, and a standard trailer box of 8.3 ft (2.53 m) internal width equals just over 500 pounds per running foot of deck. Weight per running length of deck is an intuitive unit that is understood by the audience targeted by the regulation, the truck operator who is balancing axle weight restrictions with animal needs.

Pigs can die of overcrowding in transit. Presumably, there is an empirical maximal trailer deck pressure that would result in the death of one or more pigs in 50% of the compartments loaded on trips over 8 h. Eight hours is the EU break point between short and long transport times [17]. The discovery of this standard can be accomplished by data collection at live animal receipt in a large slaughterhouse complex. The establishment of an LP_50_ for market-weight pigs in transit would allow both a regulatory standard and easy monitoring at the abattoir. Documentation of an LP_50_, a science-confirmed metric, would be an excellent performance-based trigger to initiate corrective action and to fulfill the public mandate for enforcement. The US alone slaughters 128 million pigs per year that arrive at the abattoir in loads of around 220 pigs. People loading pigs have an error rate at some level, providing detectable signals at unloading, specifically DOA and fatigued pigs.

Alternatively, axle weight limitations and trailer construction practice in a jurisdiction could effectively minimize any risk for overloading in pig transport. Specialized trailers for transporting sheep have four decks, as sheep are not as efficient to transport as pigs. Canadian transporters may operationally reserve their long trailers with increased axle weight capacity for moving cattle after removing the middle deck, as larger cattle are more efficient to transport than pigs.

In practical abattoir inspection, data captured from all the truck compartments when one or more animals are dead/fatigued on arrival could provide accumulated data to clearly identify overcrowding as a contributor to animal suffering or as presenting no risk, as apparently is the case in compliant EU jurisdictions. Establishing an LP_50_ for market pigs could be a straightforward abattoir data capture project without resorting to experimental trials or ethical animal use review committees. 

## 10. Conclusions

There is currently insufficient public oversight and the absence of a good law in the regulation of space allowance for pigs in transit. Animal welfare in agriculture is increasingly becoming part of bilateral trade negotiations [60], and the national veterinary infrastructure is failing if it is unable to respond to emerging phytosanitation and methods for production certification for international trade.

The scientific community should renew their interest in controlling the risk of overcrowding of livestock in transport in North America. European regulatory authorities may want to review the social license to maintain the current standard. An unnecessarily restrictive standard has negative implications for environmental protection and cost of production and does not increase the welfare of pigs over that of a correct efficient standard. Overregulation also endangers the legitimacy of the oversight agency.

The scientific community should harmonize the discourse on and unit of measurement of “crowding” for livestock in transit to allow for clear communications and the development of transparent regulatory standards.

This article asserts there is an LP_50_ for market pigs in transit and that it is approximately 300 kgm^−2^. EU and UK pig transporters already compliant with the 2005 maximum deck pressure convention are currently unable to provide input to this question. Pork producers in currently more unregulated jurisdictions should seek this performance-based standard before some other less efficient rule is implemented by regulatory experimentation.

In the everlasting battle against animal cruelty, the regulatory infrastructure must show courage and innovation. Courage is required, because most regulatory initiatives are experimental in nature and trigger criticism from the potentially regulated. Regulations by their nature must be implemented to identify their consequences, both hoped for and unintended. Innovation can be an outcome of regulation, as demonstrated in the vehicle speed limit initiative. The author believes that the graphical standard of maximal deck pressure is entirely coherent, meets the definition of relatively simple data in graphs [57], and is implementable in the livestock industry. The only way to identify if this belief is in error is to try. Lack of a regulatory standard impairs the public duty of regulatory bodies, does not seem to be a sustainable approach, and may allow ongoing harm to animals.

## Figures and Tables

**Figure 1 animals-14-02732-f001:**
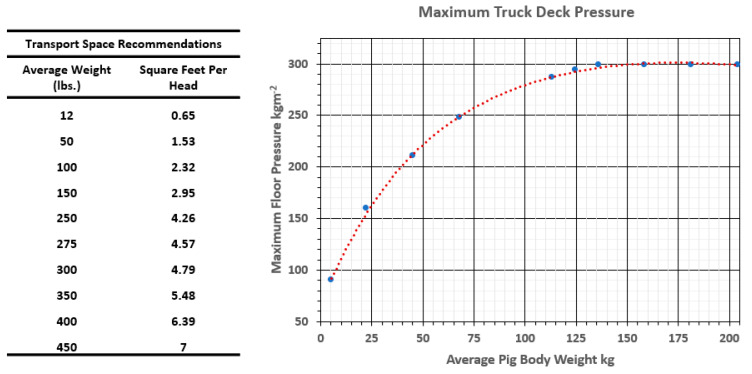
Presentation of a body weight–space required recommendation in tabular format to the same recommendation presented as a body weight vs deck pressure maximum for transporting all weights of pigs. National Pork Board space in trailer table (left) [19] (p. 27) and the same data in a maximum floor pressure graph (right). The table is a reproduction of part of the truck loading recommendations by the NPB TQA humane transport program (original table goes to 550 lb). Each line in the table gives a mass and an area, the units required to calculate pressure. The blue points on the graph to the right are the average weight (lb) divided by the square feet per head, giving units of pressure, mass/area. The pressure value converted into metric units is graphed against the body weight for each line in the table. The small red dots are the result of a curve-fitting program within the graphics options in MS Excel (smoothing of last 3 weight recommendations). The table gives 8 useful pieces of information relating average body weight and maximum deck pressure (flat after 150 kg), while the graph provides infinite information relating live individual body mass to safe maximum deck pressure. Recommendations presented as weight–space tables can be checked for internal agreement by transforming them into a pressure graph and examining goodness of fit using any curve-fitting software. The internal agreement for the first six points in this table is perfect, considering the recommendation is truncated at two decimal points. This level of precision suggests that the table’s weight range of 12–450 pounds (5–204 kg) was generated from a formula, although no reference or formula is given and heavier weights diverge from a smooth line and show more scatter.

**Figure 2 animals-14-02732-f002:**
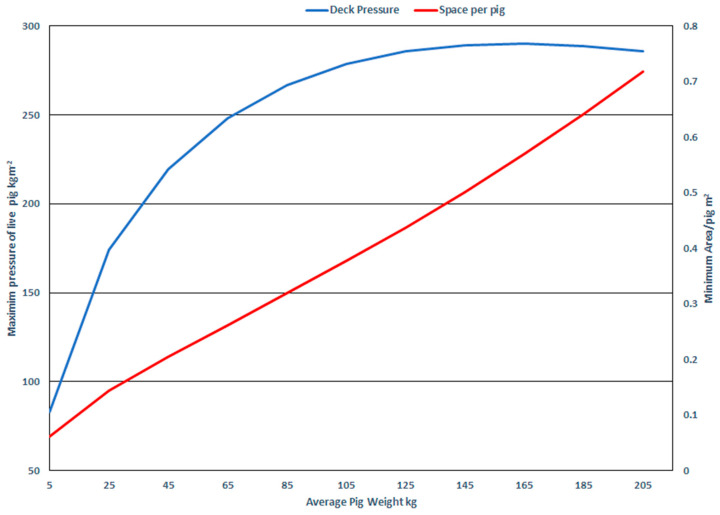
Graphical presentation of animal crowding data by two measures. The lines are both generated by the CARC prediction formula P = (37.53)(0.9969)^W^(W^0.5008^), where P is trailer floor pressure in kgm^−2^ and W is average pig weight in kg [21]. The x-axis is the average weight of a pig in a group. Floor pressure, the blue line and read against the left axis, is equivalent to the space allowance, which is represented by the red line, read against the right axis for each body weight. The deck pressure representation clearly indicates that in the biological (mathematical) relationship between body weight and space required, there is a maximal floor pressure after pigs reach around 140 kg body weight. That biological limit cannot be intuitively recognized from the red line, showing increasing space allowance per pig as average body weight increases.

**Figure 4 animals-14-02732-f004:**
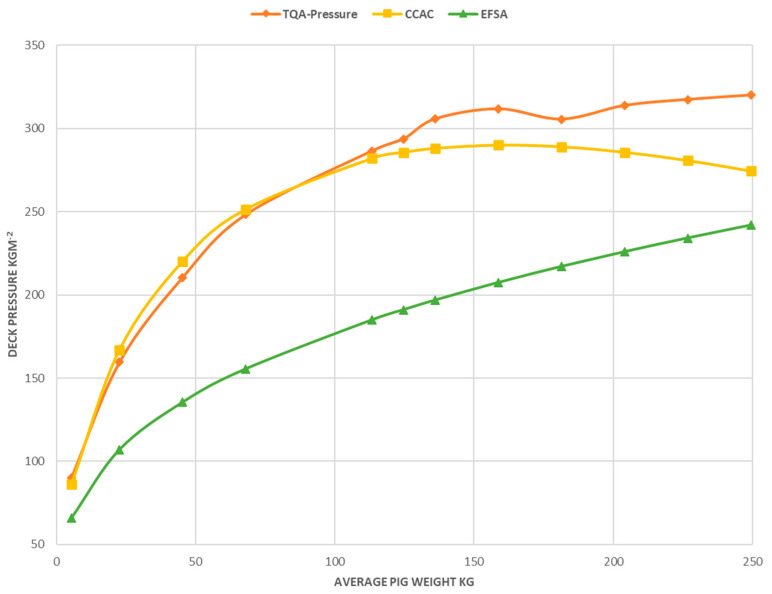
Graphical representation of maximal floor pressure recommendations for group-loaded pigs in transit from 5.5 to 250 kg from three sources. The green line is from the EFSA recommendation (Area = 0.027XWeight^2/3^) [13] (p. 88) and converted into equivalent truck deck pressure. The red line (TQA (total quality assurance) Pressure) is from the National Pork Board (US). Recommendations are given in the table [19] (p. 27), and for this graphic, converted to maximum metric pressure units from transport space recommendations. The yellow line is the CARC recommendation prediction formula in the Canadian Agri-Food Research Council maximum floor pressure for pigs in transit 2001 [18], P = (37.53)(0.9969)^W^(W^0.5008^), where P is trailer floor pressure in kgm^−2^ and W is average pig weight in kg. This is a Hoerl formula in the power-law family generated from curve fitting to observational data [21]. The markers are at irregular body weight intervals as the tabular data from the National Pork Board (NPB) recommendations have variations in body weight intervals (see Figure 1).

**Figure 5 animals-14-02732-f005:**
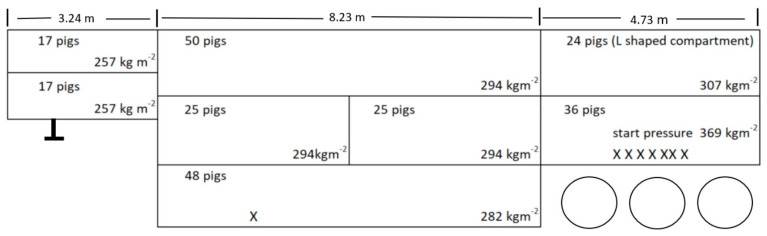
Location of dead pigs (X) identified on a warm summer day after travelling at least 16 h in western Canada. The bottom rear compartment was loaded at 369 kgm^−2^ and pigs died until the floor pressure of live pigs had dropped to 297 kgm^−2^. No deaths occurred in four central compartments at a floor pressure of around 295 kgm^−2^. The lengths of trailer compartments are above the image. Standard Canada–US–Mexico livestock trailers have an internal width of 2.53 m.

**Table 1 animals-14-02732-t001:** Evolution of national humane livestock transportation law in Canada: Health of Animals Regulation Part XII.

19 February 2020	20 February 2020—Present
Prohibition of Overcrowding	Overcrowding
140 (1) No person shall load or cause to be loaded any animal in any railway car, motor vehicle, aircraft, vessel, crate or container if, by so loading, that railway car, motor vehicle, aircraft, vessel, crate or container is crowded to such an extent as to be likely to cause injury or undue suffering to any animal therein.	148 (1) No person shall load an animal, or cause one to be loaded, in a conveyance or container, other than a container that is used to transport an animal in an aircraft, in a manner that would result in the conveyance or container becoming overcrowded, or transport or confine an animal in a conveyance or container, or cause one to be transported or confined, in a conveyance or container that is overcrowded.
(2) No person shall transport or cause to be transported any animal in any railway car, motor vehicle, aircraft, vessel, crate or container that is crowded to such an extent as to be likely to cause injury or undue suffering to any animal therein.	(2) For the purposes of subsection (1), overcrowding occurs when, due to the number of animals in the container or conveyance, (a) the animal cannot maintain its preferred position or adjust its body position in order to protect itself from injuries or avoid being crushed or trampled; (b) the animal is likely to develop a pathological condition such as hyperthermia, hypothermia, or frostbite; or (c) the animal is likely to suffer, sustain an injury or die.

## Data Availability

Data are contained within the article are from personal case notes of the author.

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
