# Peer review of "Minimum Space When Transporting Pigs: Where Is the “Good” Law?"

_animals, 2024, doi:10.3390/ani14182732_

Round 1

Reviewer 1 Report

Comments and Suggestions for Authors

In the article, it would be worth clearly formulating the purpose of the research/research study undertaken. It would be worth writing the sentence: The purpose of the research/research study was … Regardless of this, it would be worth writing what was the cognitive (scientific) purpose of the research and what was the utilitarian (practical) purpose of the research study undertaken by the Author.

Based on the review of the state of knowledge, it would be worth formulating a research problem. You could write: “The research problem is …”. The research problem can be linked to the presentation of a gap in the current state of knowledge and the research area under consideration. The information needed to formulate the research problem and identify the gap in the current state of knowledge was mostly presented in the initial part of the article. It just needs to be formulated appropriately in the body of the article. At the end of the first paragraph, the Author wrote about the lack of scientific progress … This lack is the gap in current knowledge, it just needs to be called a gap.

I do not understand why the Author uses the kgm-1 unit in lines 212-217. The correct kgm-2 unit was included in the earlier and later parts of the paragraph. The next paragraph uses the kgm-2 unit, which is correct. Therefore, it is worth correcting or explaining the kgm-1 unit.

If so much attention was paid to the floor in the study (taking into account the connections between the mass of animals and the floor surface), it may be worth developing additional issues regarding the quality of the floor. The density of animals on the surface is certainly important, but so is the quality of this surface. In my opinion, it would be worth developing in the article the issues of the impact of the quality of the floor (e.g. anti-slip properties, hardness, bedding material coverage, etc.) on the welfare of transported animals (pigs).

Does the "k" coefficient (line: 243 and subsequent lines) have a name? It may be worth including the name of the "k" coefficient in the description presented in the paragraph. In addition, it would be worth writing whether "k" is an empirical coefficient (determined on the basis of research) or a model coefficient, on what basis it was determined and how it is selected for calculations.

The length3 and length2 entries (lines: 274-275) are not fully understandable to me. Is it length cubed and squared? If so, then you need to write it differently or add a comment/explanation so that the notation is not ambiguous.

It seems to me that in the description/caption to Figure 2 it would be worth providing data on the limitations in the interpretation of the changes (curves) under consideration. I mean the group of animals, or rather the weight of individual animals (pigs), for which this graph is valid. If we take into account pigs weighing 100-110 kg each and pigs weighing, for example, 20-30 kg each, will the graph (and the range of data on the abscissas and ordinates) look the same? I am afraid it will not. Therefore, it is necessary to write for which group of pigs (with what individual weight of animals) the course of the curves in Figure 2 is valid.

If the Author presents the results of his own research (in the Case Report chapter), it would be worth presenting at least an outline of the research methodology. In the considerations presented in the article, the Author basically never used the word method/methodology. In an article presenting the results of research, one can expect the inclusion of the phrase Material and Method of research.

If the article suggests introducing the LP50 indicator into practical use, it would be worth writing more about the basis for this indicator. In the article, the LP50 indicator was mentioned only in the Abstract, in the last paragraph (before the Conclusions section) and in the Conclusions. In my opinion, it would be worth writing more about the theoretical/practical assumptions that led to the generation of the LP50 indicator. You can also write about the imperfections of the previous indicators that justify the introduction of the LP50 indicator.

If the Author makes a citation, for example Warriss (1998) (line 200), it would be better to write Warriss [19], because then it is easier to identify the given citation in References. This remark also applies to many other citations made in this way.

It would be worth reviewing the article for minor spelling inaccuracies. For example, on line 269 (caption under Figure 2) there are two "is" next to each other.

Author Response

Comment Reviewer 1

Response

In the article, it would be worth clearly formulating the purpose of the research/research study undertaken. It would be worth writing the sentence: The purpose of the research/research study was … Regardless of this, it would be worth writing what was the cognitive (scientific) purpose of the research and what was the utilitarian (practical) purpose of the research study undertaken by the Author.

Good Call now lines 66-72

Based on the review of the state of knowledge, it would be worth formulating a research problem. You could write: “The research problem is …”. The research problem can be linked to the presentation of a gap in the current state of knowledge and the research area under consideration. The information needed to formulate the research problem and identify the gap in the current state of knowledge was mostly presented in the initial part of the article. It just needs to be formulated appropriately in the body of the article. At the end of the first paragraph, the Author wrote about the lack of scientific progress … This lack is the gap in current knowledge, it just needs to be called a gap.

New Sentence line 47

I think I have answered this on lines 61-67

I do not understand why the Author uses the kgm-1 unit in lines 212-217. The correct kgm-2 unit was included in the earlier and later parts of the paragraph. The next paragraph uses the kgm-2 unit, which is correct. Therefore, it is worth correcting or explaining the kgm-1 unit.

Clearly my error thanks.

Corrected error now on lines 230-247

If so much attention was paid to the floor in the study (taking into account the connections between the mass of animals and the floor surface), it may be worth developing additional issues regarding the quality of the floor. The density of animals on the surface is certainly important, but so is the quality of this surface. In my opinion, it would be worth developing in the article the issues of the impact of the quality of the floor (e.g. anti-slip properties, hardness, bedding material coverage, etc.) on the welfare of transported animals (pigs).

In the situation I see of overcrowding the nature of the floor is not highly relevant. The crowding exceeds comfortable standing room only threshold and pigs fight for floorspace to lie on.

My interest is not in pig comfort, it is in identifying cruelty to animals caused by overcrowding and brining the matter before the judicial arm of government. What I need is the law to describe an offence in a manner that I can convince a judge that animals suffered.

Does the "k" coefficient (line: 243 and subsequent lines) have a name? It may be worth including the name of the "k" coefficient in the description presented in the paragraph. In addition, it would be worth writing whether "k" is an empirical coefficient (determined on the basis of research) or a model coefficient, on what basis it was determined and how it is selected for calculations.

Thank you for this comment I did not know how much information I needed to include to identify this principle. I have expanded this section and included 2 new references specifically on scaling.

And a couple of new sentences about pigs tend to maintain heart function to body mass ratio (Scaling) up to at least 75 kg (van Essen 2009) while adult breeding sows have much poorer proportional cardio performance.

This better completes the thread of the scaling discussion in the paper.    See lines 267-276

The length3 and length2 entries (lines: 274-275) are not fully understandable to me. Is it length cubed and squared? If so, then you need to write it differently or add a comment/explanation so that the notation is not ambiguous.

Corrections lines 312-322

I changed figure 2 completely so it is less confusing and less reliant on the scaling principal.

It seems to me that in the description/caption to Figure 2 it would be worth providing data on the limitations in the interpretation of the changes (curves) under consideration. I mean the group of animals, or rather the weight of individual animals (pigs), for which this graph is valid. If we take into account pigs weighing 100-110 kg each and pigs weighing, for example, 20-30 kg each, will the graph (and the range of data on the abscissas and ordinates) look the same? I am afraid it will not. Therefore, it is necessary to write for which group of pigs (with what individual weight of animals) the course of the curves in Figure 2 is valid.

Expanded the caption to the Figure 2

If the Author presents the results of his own research (in the Case Report chapter), it would be worth presenting at least an outline of the research methodology. In the considerations presented in the article, the Author basically never used the word method/methodology. In an article presenting the results of research, one can expect the inclusion of the phrase Material and Method of research.

Included 2 sentences at 403-406

If the article suggests introducing the LP50 indicator into practical use, it would be worth writing more about the basis for this indicator. In the article, the LP50 indicator was mentioned only in the Abstract, in the last paragraph (before the Conclusions section) and in the Conclusions. In my opinion, it would be worth writing more about the theoretical/practical assumptions that led to the generation of the LP50 indicator. You can also write about the imperfections of the previous indicators that justify the introduction of the LP50 indicator.

Added line 540-542

Line 549-544

If the Author makes a citation, for example Warriss (1998) (line 200), it would be better to write Warriss [19], because then it is easier to identify the given citation in References. This remark also applies to many other citations made in this way

Changes Made

It would be worth reviewing the article for minor spelling inaccuracies. For example, on line 269 (caption under Figure 2) there are two "is" next to each other.

Changes Made

Reviewer 2 Report

Comments and Suggestions for Authors

  This is a polemic against current rules on swine transport. It is not a research paper.   The English is awkwartd65 this sentence does not make sense; perhaps  avoidance of prohibited behavior and compliance...

68 voluntarily 

72 what should be added ;performance based?

79 what is a prescriptive law?

84 define bright line   84 define bright line   

135 on Tbale or in text indicated that the left side is the old regulation and the right side the now one/

138 Why is it difficult. Do you mean there was a great improvement?

153 conflict between various rules

155 Do you mean  the high turn over rate, poor pay etc. and a hazard? 

192 please explain why 4500  kg can't be utilized

199 do you mean serological or physiological ( cortisol, for example .

206 meat quality deterioration 

216  &230give Warriss year again

265 needlessly conservative  Do you mean more pr less space/p ig needed

275 imagineries???  Do you mean calculations  or hypothetical pressures Also there are three lines not two.

283 omit comma 

286 and 328  bloom of youth is too informal  growth spurt perhaps 

312 the authior's personal preference.

294 not enjoy where  veterinary inspection    is required at the border

Fig 3 until the floor pressure of live pigs had dropped to 297 kgm- Is that really trhe reason no more died. ( ie if enough pigs die the survivors have good welfare?

371 worst offender vile is not scientific 

384 commas would help  convinced, defense,

391 suddenly  a different topic is raised( frost bite )rather than crowding )

445 Are you suggesting re-writing the legislation with more understandable

453terminology? 

414 there is 

433 omit wicked.  Are you implying that truckers understand th rules but the enforcers ( lawyer and authors of legislation)  of th rules do not?

453People loading pigs have an error rate at some level, providing detectible signals at unloading, DOA and stressed  What do you mean are the errors significant? 

464 phytosanitary??? clean plants 

 477" should be seeking" for    should be "should seek"

Comments on the Quality of English Language

Included in comments to author 

Author Response

Comment Reviewer 2

Response

  This is a polemic against current rules on swine transport. It is not a research paper.   The English is awkwartd65 this sentence does not make sense; perhaps  avoidance of prohibited behavior and compliance...

Replaced with Regulatory Compliance Now line 77

68 voluntarily

In the example of speeding in the next paragraph people voluntarily comply or do not comply with speeding laws Now line 75

Also 2 new sentences and new reference at Line 76-77.

72 what should be added ;performance based?

79 what is a prescriptive law?

New sentence and reference Now lines 86-93

84 define bright line   84 define bright line 

Removed the term bright line – it is from common law where subsequent case ruling eventually clarifies a poorly written statute.

It qualifies as jargon in this application Good comment from reviewer

Replaced with legitimate and offered a reference for this legal-policing concept of legitimate law.  Now line 104-105

135 on Table or in text indicated that the left side is the old regulation and the right side the now one/

This is correct

138 Why is it difficult. Do you mean there was a great improvement?

Added a sentence Now line 173-179

153 conflict between various rules

Added a clause New sentence 178-179

155 Do you mean  the high turn over rate, poor pay etc. and a hazard?

Added clarifying clause Now line 178-179

192 please explain why 4500  kg can't be utilized

I have redrawn figure 1 and this is a promotional source for compartment size see reference (1) I am reluctant to include promotional material in a publication.  although in the Case Study The trailer was measured New paragraph line 206-220

On new line 201 I changed possum to pot as both are jargon and “pot” is used in (2).

See new text Lines

199 do you mean serological or physiological ( cortisol, for example

No changes made in this sentence

206 meat quality deterioration

216 &230give Warriss year again

Corrected

Added a new paragraph on line 326-334 to clarify the intent of figure 2 3 new references van Essen

265 needlessly conservative  Do you mean more pr less space/p ig needed

New Figure and mostly new caption.

275 imagineries???  Do you mean calculations  or hypothetical pressures Also there are three lines not two.

Replaced with “mental constructs” now line 292

283 omit comma

Now line 300

286 and 328  bloom of youth is too informal  growth spurt perhaps

Reworded line

312 the author’s personal preference.

I don’t think this statement is confusing

Added a sentence at  line 386-372

Clarify that preference is not to be confused with whim.

Included a new figure 4 as this is a point is really one of pedagogical utility not a “preference”.

294 not enjoy where  veterinary inspection    is required at the border

Originally 394 now line 336 There is veterinary border inspection

Fig 3 until the floor pressure of live pigs had dropped to 297 kgm- Is that really trhe reason no more died. ( ie if enough pigs die the survivors have good welfare?

Now Fig 4

New Paragraph Line 432-437

371 worst offender vile is not scientific

New wording line 427

384 commas would help  convinced, defense,

New wording 439

391 suddenly  a different topic is raised( frost bite )rather than crowding )

Frostbite is in the prohibition of overcrowding section of the statute. In Canada you have to decrease the floor pressure so that pigs can easily move away from cold spots in the truck. If it is too much work to move a pig will lie in a cold spot and suffer frostbite.  This is what the law is referring to, but no prosecutor would have a clue how to describe this to a judge. Just an embarrassment of a regulatory wording.

New paragraph line 469-477

445 Are you suggesting re-writing the legislation with more understandable

Yes I am. Now line 521-526

453terminology?

414 there is

433 omit wicked.  Are you implying that truckers understand th rules but the enforcers ( lawyer and authors of legislation)  of th rules do not?

Now line 521-526. Added a paragraph added a wicked problem reference if that was the concern and more specifically described the numeracy problem

453People loading pigs have an error rate at some level, providing detectible signals at unloading, DOA and stressed  What do you mean are the errors significant?

464 phytosanitary??? clean plants

The Agreement on the Application of Sanitary and Phytosanitary Measures (the "SPS Agreement") entered into force with the establishment of the World Trade Organization on 1 January 1995. It concerns the application of food safety and animal and plant health regulations. https://www.wto.org/english/tratop_e/sps_e/spsund_e.htm

477" should be seeking" for    should be "should seek"

Changed Line 540

Round 2

Reviewer 2 Report

Comments and Suggestions for Authors

 Axle is misspelled throughout as axel 

191The adoption of weigh-in-motion technology by enforcement agencies (19) may further assure compliance with this directive is dominant over intersectional risks inherent in adjusting livestock compartment floor pressure to assure preferred center of gravity of the tractor trailer system.

" compliance with this directive is dominant over intersectional risks"  does not make sense

224  Warriss give year

249 give year

Figure 2  What do orange and green lines meangive year ( I asked for this correction previously )

496 It is not clear the origin or complete nature 

of this problem. Should be The origin or complete nature of the problem is not clear 4

of this problem.

499 are suspected or in fact do not 

500 there is  a numeracy problem"may not understand'

519  serious not wickwd 

Comments on the Quality of English Language

comments are in suggestions to author 

Author Response

Round 2 Review

Comment

Reply

Axle is misspelled throughout as axel

Correction Made X28

191 The adoption of weigh-in-motion technology by enforcement agencies (19) may further assure compliance with this directive is dominant over intersectional risks inherent in adjusting livestock compartment floor pressure to assure preferred center of gravity of the tractor trailer system.

Inserted a new sentence at line 267

Replaced problematic phrase with regulation preventing livestock overcrowding where detection is unlikely unless there is animal death in transit.

" compliance with this directive is dominant over intersectional risks"  does not make sense

See above 196 to 204   

Curve fitting

224  Warriss give year  249 give year

Change made  Now line 297

Figure 2  What do orange and green lines mean give year ( I asked for this correction previously )

Line 305 to 315 Repeated in new Figure 3,

Wrote prediction formulae into the legend

496 It is not clear the origin or complete nature of this problem. Should be The origin or complete nature of the problem is not clear 4 of this problem.

Introduced a new figure 1 that gives instruction and an example of representing BW-space recommendations in a table and the option of co9mmunicating the same recommendation in a BW-pressure graphic

499 are suspected or in fact do not

Paragraph starting Line 523

500 there is  a numeracy problem "may not understand' 519  serious not wicked

Corrected the reference for “wicked problem” removed  and the phrase wicked problem removed from the document
